# Research on the Frontier and Prospect of Service Robots in the Tourism and Hospitality Industry Based on International Core Journals: A Review

**DOI:** 10.3390/bs13070560

**Published:** 2023-07-05

**Authors:** Mengxi Chen, Xiaoyu Wang, Rob Law, Mu Zhang

**Affiliations:** 1Department of Tourism Management, South China University of Technology, Guangzhou 511442, China; td_chan@mail.scut.edu.cn; 2Department of Tourism Management, School of Management, Jinan University, Guangzhou 510632, China; centyu@stu2021.jnu.edu.cn; 3Department of Integrated Resort and Tourism Management, Faculty of Business Administration, University of Macau, Macau 519000, China; roblaw@um.edu.mo; 4Asia-Pacific Academy of Economics and Management, University of Macau, Macau 519000, China; 5Shenzhen Tourism College, Jinan University, Shenzhen 518053, China

**Keywords:** service robots, tourism and hospitality industry, web of science, bibliometric analysis, service-dominant logic

## Abstract

This paper used the mixed research method of bibliometric and content analysis to study 284 studies on service robots in the tourism and hospitality industry collected from the Web of Science database. Results show that research in this field started late, and that the COVID-19 pandemic has promoted the rapid growth of the number of research papers. The *International Journal of Contemporary Hospitality Management* has so far published the most number of papers. Numerous scholars from universities in different regions of the world have made significant contributions to the research of service robots, and academic collaborations are relatively common, but there are only very few high-yield authors. Empirical research has been widely favored by researchers, wherein online questionnaire and experimental methods have been frequently used. Multidisciplinary theories have also been cited in related articles, especially on the applications of psychological theories. The research fronts cover four branches focusing on service robots, consumers, human employees, and service environment, with all four parts largely overlapping in content. Finally, the paper discusses prospects for the future research agenda of service robots in the tourism and hospitality industry.

## 1. Introduction

Over the past decade, advances and applications of robotics-related technologies have changed the way products and services are delivered due to advantages such as efficiency, cost savings, and the standardization of services [1]. The tourism and hospitality industry is an important part of the service industry, aiming at providing transportation, accommodation, and entertainment services for the public [2], including subsectors such as lodging, attractions, transportation, food services, events, and time share [3,4]. Compared with other industries, the emphasis on service experience and interaction has resulted in service robots to possess broad application prospects in the tourism and hospitality industry [5]. Robots are used in tourism and hospitality service practices to provide services such as information consultation, service delivery, concierge, and room cleaning [6]. Some destinations and large hotels even introduced artificial intelligence (AI)-enabled robots to provide more personalized services and new experiences to consumers. Due to the advantages of contactless services, robots have become more popular after the outbreak of the coronavirus disease 2019 (COVID-19); when the public began to attach more importance to self-health and safety [7], services such as facial recognition, automatic disinfection, and auto check-ins were performed by robots [8]. The benefits of service robots to the tourism and hospitality industry are undeniable, but there are also potential risks. In addition to privacy concerns and threats to human identity [9], one of the most urgent issues is the replacement of human labor. The McKinsey Global Institute predicted that by 2030, at least one-third of the world’s sixty percent of occupations could be replaced by automated technologies like robots, and up to 800 million people worldwide will be affected [10], suggesting that the application of service robots may cause permanent unemployment in the long run [11]. It is inevitable that the tourism and hospitality industry will experience remarkable changes in accordance with the influx of service robotic technology within the foreseeable future.

The impact caused by service robots has further sparked academic attention. A major increase in studies examining the application of service robots in tourism and hospitality can be seen in recent years, mainly focusing on its impact on consumers, employees, and other stakeholders [12]. Some scholars attempted to display the findings of existing research from different perspectives, such as the review of research on human–robot interaction [5], the summary of the factors that affect consumers’ use of robotic technology [1], as well as the generalization of marketing factors influencing the introduction of service robots in the service industry [13]. However, most of the review studies are descriptive and conceptual, mainly focusing on a single topic. Although several literature reviews claimed to have comprehensively analyzed the academic gaps of service robots in the tourism and hospitality industry [12,14], the research on service robots has increased by a substantial amount after the outbreak of COVID-19, alongside some new research issues. It is expected that the application of service robots in the tourism and hospitality industry will further rise in the future [1]. Therefore, it is significant to organize the existing research on service robots in the tourism and hospitality industry, not only to understand the changes service robotic technology have brought into the industry, but also to understand the latest academic progress on this topic, so as to explore possible research gaps. At present, the amount of scientific literature on service robots is still limited, but from a practical application point of view, the growth potential of this domain in scholarly writings is extensive. Hence, a comprehensive and systematic review of existing research on service robots in the tourism and hospitality industry is necessary in order to provide relatively up-to-date and valuable insights for future research. Given this, we try to resolve the following three questions:

Q1: What is the most basic and latest situation of service robotics research in tourism and hospitality? 

Q2: What are the research fronts of service robots in the tourism and hospitality industry? 

Q3: What are the existing research gaps in service robotics in the tourism and hospitality industry?

Accordingly, this study uses the literature of service robotics research in tourism and hospitality in the core database of Web of Science (WoS) as research data to provide a relatively complete summary of the existing research. Considering the “service-oriented” nature of the tourism and hospitality industry [9], service-dominant logic (S-D logic) is used as the research basis. S-D logic, first proposed by Vargo and Lusch [15], is a theoretical framework in the field of marketing and management. This theory focuses on service exchange and value co-creation, emphasizing the service participation and value contribution of multiple participants [16]. The adoption of service robots in tourism and hospitality means that service robots are involved in the service delivery system in the industry; S-D logic is therefore a useful theoretical perspective to understand the process of value creation associated with robotics in hospitality and tourism [9]. A mixed research method combining bibliometric and content analysis was used. The bibliometric analysis method was applied to statistically analyze the quantitative information of existing research papers, while content analysis was used to explore the qualitative content such as research topics on service robots in-depth. Based on the results of the analysis, we then provide a research agenda for future research.

This paper has several major contributions. First, the usage of mixed research methods provides a holistic and objective review of the previous literature, which can reduce the potential subjectivity bias of a single approach. Second, by reviewing the existing literature on service robots in tourism and hospitality, the paper integrates and synthesizes extant knowledge to provide a state-of-the-art understanding of this research field. Furthermore, the findings will be useful for scholars to identify the most influential papers, authors, and institutions in this academic area, as well as have a general understanding of the frontier of service robotics research, which may help them discern the potential research gaps. In addition, this study is the first of a few to use S-D logic to carry out the review research, where we examine the hosts (service robots and employees), customers, service environments, and relationships between them, all of which play an important role in service delivery systems.

## 2. Definition of Service Robots in the Tourism and Hospitality Industry

From existing research, there is no strict and uniform definition of service robots, and the views of researchers vary from country to country. The International Organization for Standardization has defined a service robot as a “programmed actuated mechanism used in personal or professional applications that performs useful tasks for humans or equipment” [17]. The definition of service robots has since expanded as research has progressed and diverged in content. Wirtz et al. defined the front-line service robot as a robot designed on a system basis to be autonomous and flexible, and used to interact, communicate, and provide services to customers [18]. This point of view highlights the difference between service robots and self-service technology, that is, service robots are more flexible and adaptable to their environment and have machine learning capabilities. A more widely accepted definition comes from the International Federation of Robotics, which argues that “a service robot is an actuated mechanism programmable in two or more axes, moving within its environment, to perform useful tasks for humans or equipment excluding industrial automation applications” [19] (p. 20). This definition further emphasizes the autonomous, non-productive, and value-based features of service robots. 

Some studies have defined service robots without making a strict distinction between them and AI, but the two are fundamentally different: AI focuses on the technical ability to perform tasks, mostly involving programs and algorithms, generally without physical entities, and usually as text- or voice-driven agents [13]. The connection between robot and AI is that robots are one of the key vehicles for AI technology and both have automated characteristics, with AI being an important means of enhancing the intelligence of robots, enabling them to have human-like intelligence such as learning and reasoning [20]. However, the difference between the two is also clear, namely that when AI is not involved, robots focus on completing mechanical tasks, without having AI’s ability to learn perception, reasoning, and actuation to replace human intelligence. Moreover, the implementation of AI mainly focuses on programs and algorithms, while the design of robots also pays attention to the hardware-like physical appearances [13]. Based on this, the paper argues that tourism and hospitality service robots (hereinafter service robots) are non-productive machines that are integrated by various types of high technology, can autonomously perform physical or non-physical tasks, and communicate, interact, and provide customized services to customers.

## 3. Methodology

### 3.1. Data Collection and Cleansing

WoS was chosen as the data pool because the database is highly influential and contains more than 10,000 peer-reviewed high-quality social science and humanities journals [21]. Most importantly, unlike other databases (e.g., Scopus), WoS identifies tourism and hospitality as an independent academic category [22], which is more in line with our research context. We conducted the literature searches in the core database of WoS during 5–10 November 2022 and 13–17 April 2023, respectively, each consisting of two-round data cleansing processes to extensively retrieve the relevant literature. 

To ensure the accuracy of the search string, we referred to the work of previous studies [1,3,4] to classify the subsectors of the tourism and hospitality industry. Before performing the search, two experts in the field of tourism and hospitality were invited to review the preliminary search string. After considering their suggestions, the final search steps were as follows. First, we used the subject term search function on WoS with the keyword set (“topic = service robot” OR “service robotic”) AND (“subtopic = tourism” OR “hospitality” OR “hotel” OR “restaurant” OR “destination” OR “transport” OR “event” OR “club” OR “leisure”). Second, “English” was selected as the main language because it is the most widely spoken language in the world, and the time period was unlimited. In addition, in order to prevent any missing literature concerning service robots as AI technology, we also reviewed the abstract of the papers from the references of the literature obtained through keyword search to supplement our dataset by snowballing approach, which is a useful methodology in exploratory and qualitative research, with advantages in terms of expanding sample size and reducing costs and time [23]. The preliminary search yielded 1847 and 595 documents over two time periods, respectively.

Duplicate and incomplete studies were removed in the first round of data filtering. In the second stage of data cleansing, we closely reviewed each paper and judged the relevance of the topic to service robots, further excluding papers that only mentioned service robots in the limited section and papers which studied non-hospitality and tourism contexts. Moreover, only peer-reviewed journals were considered for the review in order to focus on the representativeness and quality of the data [24], meaning book reviews, discussions, opinions, and conference proceedings are not included. In this stage, 241 and 43 papers were retrieved, respectively. Finally, a total of 284 potential studies were obtained, covering the period from 2012 to 2023.

### 3.2. Data Analysis

The research data were processed using Microsoft Excel and CiteSpace 6.2, both of which are freely accessible and available to the public. Excel was chosen to organize the basic information of the retrieved literature. While CiteSpace, a bibliometric analysis program created by Chen [25], which is commonly used by researchers to explore the knowledge base, hotspots, and fronts of a research topic [26], was used with the keyword cluster analysis function to lay the groundwork for subsequent content analysis. This was because the keywords of the literature are highly concentrated, which is considered to reflect the main idea of the research [26], and keyword cluster analysis is a bibliometric approach to group keywords into a set on the basis of similarity and dissimilarity, having advantages in recognizing the same research interests or themes [27]. This paper attempts to explore the research fronts of service robotics research in the tourism and hospitality industry via keyword cluster analysis. The steps of the data analysis are as follows.

First, we examined several highly cited review studies on tourism and hospitality research in WoS, from which the framework of the current research analysis was extracted, including literature types, years of publication, journal sources, researchers, research institutions, regions, and research methodology, and theory. Second, the retrieved literature was sorted and analyzed following the predetermined framework. The basic structure of the research topic was then obtained through keyword cluster analysis. To avoid the unilateralism of the mechanical software analysis, the content analysis method was combined with an in-depth analysis of the relevant literature content to optimize and re-code the results of the keyword clustering, and research fronts were further analyzed based on the encoding results. Finally, the research gaps were unearthed according to the findings obtained.

## 4. General Findings

### 4.1. Trends in the Current Literature

The years of publication of the 284 studies were counted and the results can be seen in Figure 1. It is clear that research on service robots first appeared in 2012, but academics only began formal explorations in 2015. This phenomenon is closely related to the application of service robots: in 2015, the first robot hotel, Henn-na Hotel, opened in Nagasaki Prefecture, Japan. Since then, service robots have successively “settled” in major hotels. The robots include Connie, a smart concierge at the Hilton Hotel, robot attendants Leo and Cleo at the Marriott International [11], and service robot Pepper of the Mandarin Oriental Hotel [28]. These developments have increased the attention on service robots then. According to the statistical results, the number of related research publications has continued to grow since 2017, especially after the outbreak of COVID-19 in late 2019. During this period, the tourism and hospitality industry has been severely hit by the impact of the epidemic. The emergence of robots has made the contactless services possible, reducing the threat of coronavirus transmission among human contacts and providing a new direction for the development of the tourism and hospitality industry. In terms of research development phase, the current service robotics research has gone through its infancy and entered an exploratory stage, with a significant increase in the number of publications and a steady growth rate. 

### 4.2. Journals of Publications

The number of retrieved journals amounted to 76 journals from different subject research areas. Because WoS has a mature set of metrics to evaluate the influence of the journals it indexed [29], we recorded key indicators of high-producing journals that have more than five publications, including Journal Citation Index Category (Category), Journal Impact Factor Quartile (Quartile) and the Impact factor for the last five years (5YIF). According to Chang et al. [29], the quality of service robotics research papers in the tourism and hospitality industry can be reflected by these indicators. As shown in Table 1, three different journal citation index categories were included: SSCI (Social Science Citation Index), SCIE (Science Citation Index Expanded), and ESCI (Emerging Sources Citations Index). SSCI (80%) journals accounted for the largest proportion, followed by SCIE (18%) and ESCI (2%). All of them are core international academic journals, with tourism journals accounting for the largest proportion, such as *International Journal of Contemporary Hospitality Management*, *International Journal of Hospitality Management*, *Annals of Tourism Research*. The average impact factor of high-producing journals is 8.238, indicating that the related research is of high theoretical and practical value. Journals in other disciplines, such as *Technology in Society* and *Computers in Human Behavior* in information technology, the *International Journal of Social Robotics* in robotics, and *Electronic Markets* in marketing, also have a high number of publications. These further show that the application of service robots in the tourism and hospitality industry has attracted research attention from multiple disciplines and has become a practical issue of significance.

### 4.3. Main Researchers and Institutions

According to the statistical results, 284 papers involve 703 researchers. We counted productive researchers with more than five publications and extracted the affiliations and regions of the authors from the authorship information accompanying these papers, so as to obtain an overview of the research institutions and their geographical distribution. The research interests of the productive authors were further obtained via their affiliated official website to identify their disciplinary background (see Table 2). It is clear that the high-producing researchers are all scholars in the research field of tourism and hospitality. The most productive author is Hwang Jinsoo, a South Korean scholar who has made numerous contributions to the field of robotics in food service and consumer behavior. Additionally, Hwang collaborated closely with Kim Heather Markham, another productive author included in Table 2, and most of their research focuses on the impact of robotic services on F&B brands. Ivanov Stanislav followed Hwang in a number of publications, with related research focusing on robotics economics and automated services, and he conducted several studies with Craig Webster. Other scholars such as Tussyadiah IisP, Gursoy Dogan, and Seyitoglu Faruk have also made significant contributions to service robotics research in the tourism and hospitality industry.

With regard to research institution, a total of 383 institutions have been involved, with higher education institutions like universities or colleges accounting for the largest parts (94%), and the remainder is for corporate entities (3%) and research institutes (3%). Sixteen research institutions published more than five papers, all of which are well-known universities all over the world (see Figure 2), especially universities in China, the United States, and South Korea. This phenomenon is positively correlated with the level of economic and technological development in these countries. The United States has been at the forefront of global development in areas such as robotics technology and business services. While South Korea has excellent manufacturing capabilities and strong competitiveness in the software sector. China’s robotic technology development and academic research are later than those of the United States and South Korea, but the Chinese government has successively enacted policies such as Made in China 2025 and the Robotics Industry Development Plan (2016–2020) in recent years, accelerating the integration and development of the service industry and robotics, as well as the pace of academic research.

It is noteworthy that academic cooperation is extremely common, with more than 90% of the papers being completed by two or more researchers and research institutions. Some researchers and institutions in the same region collaborated due to regional convenience [1,30,31,32]. While more cross-regional collaborations occurred among counties [33,34,35,36,37,38]. This is particularly true between mainland China and the United States, where more and more scholars and universities have entered the research field of service robots and have conducted a large number of collaborations [33,34,35,38].

### 4.4. Research Methodology in Publications

The 284 potential studies include 37 review papers and 247 empirical papers. Review papers are critical evaluations of previously published research, providing readers with an up-to-date understanding of the research topic [39]. While empirical papers are papers that rely on evidence acquired through the scientific method of data collection or observation [40]. Both review and empirical papers mainly involve three research methods: qualitative, quantitative, and mixed methods that combine the first two methods [39,41]. Research methods vary in the types of research data: qualitative research methods involve collecting and analyzing non-numerical data. This includes case studies, personal experiences, interviews, observational data, and visual texts [42]. In contrast, quantitative research methods focus on collecting and analyzing numerical data for statistical analysis, such as study designs, statistical methods [43] and bibliometric analysis [44]. Meanwhile, mixed research methods employ both quantitative and qualitative research techniques in either parallel or sequential phases [45].

Accordingly, the research methods covered in 284 studies can be seen in Table 3, the category of the sub-methods was inspired by the work of Khoo-Lattimore et al. [45]. Quantitative research dominated (169) as researchers try to investigate the cause–effect and correlation relationships between variables, wherein questionnaires and experimental methods were often used to collect data. Questionnaires (88) are distributed both offline and online, with more than 90% of the studies using online questionnaires due to the convenience of data collection. Experimental methods (71) included online (79%), field (11%), and laboratory experiments (10%), of which online experiments were the most commonly used due to advantages such as low cost and the absence of time and space constraints. Experimental designs were mostly simulated in real-life scenarios to enhance the persuasiveness of the experimental results, largely being accompanied by questionnaires for data collection. Simple statistics, correlation analysis, factor analysis, analysis of variance, structural equations, regression analysis, and independent sample t-tests are commonly used in the processing of sample data.

For qualitative studies (80), the literature analysis and interview were found to be the most pervasive. A literature analysis (23) usually appears together with content analysis in reviews to examine previous studies. The interview method (20) was used to explore the perception of service robots on either the demand side or supply side, and was mainly semi-structured in form, thus enabling interviewers to improvise follow-up questions based on participants’ responses, and was often combined with coding analyses to process interview data. The data mining method (17) was commonly used as well, with researchers generally using text mining tools like Python to obtain data such as online reviews, and relevant papers usually involving sentiment analysis to reveal the factors that influence users’ emotions toward service robots [46]. Moreover, a small number of studies used mixed method (35), which mostly combined interviews with questionnaires or experiments to ensure the reliability of the research results and the extensiveness of the research findings. Additionally, the processing of the research data varied depending on the type of data obtained. Generally, researchers focused on answering the research questions or hypotheses instead of innovating research methods, hence the use of methods was relatively conventional.

### 4.5. Research Theory

More than half of the studies have theoretical foundations, while the rest are mostly based on data-driven or literature analysis. Theory-driven research involves the intersection and integration of theories from disciplines such as psychology, management, sociology, information technology, marketing, and communication. Psychological theories are the most widely cited (see Figure 3), including the theory of planned behavior and cognitive appraisal theory, which are applicable to multiple research contexts, as well as theories dedicated to the study of human–robot interaction, such as anthropomorphism theory and stereotype content model. The technology–organization–environment (TOE) framework in management is also commonly used in the literature, examining the impact of robotic technology on the service supply side. Some scholars were interested in the commercial function of service robots and cited marketing theories such as consumer value theory and the service quality (SERVQUAL) model. Furthermore, the social issues raised by the application of service robots have received widespread attention as well, and sociological theories such as service role theory and social exchange theory have been used to explore the impact of robots on social relations between hosts and customers and between humans and robots in service scenarios.

The high-frequency theory (see Figure 4) shows that the technology acceptance model (TAM) is the most commonly used theory. TAM was used to study computer technology in the early days, explaining the reasons why computer technology is widely accepted by the public through the two dimensions of usefulness and ease of use, and TAM was later extended to robot research scenarios by Bröhl et al. [38]. However, it has been argued by some scholars that the traditional TAM fails to provide a complete conceptual framework for a comprehensive study of public attitudes and use of robotics, so the TAM model has been expanded and a framework for research on technology acceptance specifically for service robots has been proposed. Examples include the AI device use acceptance model (AIDUA) [47], service robot acceptance model (sRAM) [18], and interactive technology acceptance model (iTAM) [48]. The uncanny valley theory, developed from evolutionary psychology, is also regularly used, stating that the closer a robot appears and behaves to humans, the more positive emotions people are likely to feel. Yet, when the similarity reaches a peak, the higher the similarity, the more frightened people feel, forming the so-called “uncanny valley”, and as the similarity continues to rise closer to humans, people regain positive emotions [49]. The uncanny valley is often used to study the effect of robot anthropomorphism on users. Other interesting and highly cited theories include the computers are social actors (CASA) paradigm in journalism and communication, stimuli–organism–response model in cognitive psychology and affordance theory in ecological psychology, which have different perspectives but are all concerned with the social impact of service robots.

## 5. Research Frontier Exploration

Since this paper attempts to conduct an in-depth study based on S-D logic from the perspective of the service delivery system, we further organized and summarized the research frontiers of service robots in tourism and hospitality based on the S-D logic via content analysis combined with cluster analysis of CiteSpace.

### 5.1. Cluster Analysis of Keywords

By using CiteSpace 6.2 to cluster the keywords of 247 empirical studies, highly correlated keywords have been grouped into the same category, and different categories of keyword clusters have different colors [27], which is designed to intuitively reflect the general information of each cluster (e.g., cluster labels and the keywords). Accordingly, 12 clusters were obtained (see Figure 5). The values of modularity and silhouette are two key factors that explain the clustering results. Modularity is the clustering module value (often called Q-value), and it is generally believed that Q > 0.3 means that the cluster structure is significant. While silhouette (often referred to simply as S-value) refers to the average contour value of the cluster; it is generally considered that a clustering of S > 0.7 means that the clustering is convincing [25,27]. According to Figure 5, Q = 0.7637 > 0.3 and S = 0.9075 > 0.5, indicating that the clustering network structure is reasonable. Owing to the duplication and apparent deviation from the research topics of service robots in the twelve cluster labels, the following were cleansed: #4 social cognition; #5 self-service technology; and #10 experience economy, which are not strongly representative and correlated with service robots; and the repeated cluster #7 service robot. The remaining eight clusters are the basic topics of service robotics research in the tourism and hospitality industry.

### 5.2. Coding and Classification of Keywords

Considering that CiteSpace may pose problems such as improper classification and blurred clustering boundaries due to the mechanical shortcomings of the algorithm, we examined the specific content of these studies in detail and optimized and manually coded the keywords in the eight clustering blocks based on automatic clustering. Specifically, cluster #2 human–robot interaction was retained because of the direct relevance to the research on service robots. Furthermore, we renamed some cluster labels to make the connection between the clusters and service robots more intuitive, namely clusters #3 customer attributions, #6 physical appearance, #8 service quality, and #11 robotic deployment. In addition, since the labels of the clusters #0 artificial intelligence and #1 service robots are too broad, we refined these clusters through analysis. As for cluster #9 service failure, some of the keywords in this cluster are duplicated in #8, but the emerging words are closely related to the functions and roles of service robots, such as robot barista, robot concierge, and robotic chef, thus we renamed the cluster “Function and role of robots in the service context”. The results of the keywords classification optimization can be seen in Table 4, in which the clustered topics and keywords are divided into eight main categories and several subcategories.

### 5.3. Main Research Themes

Accordingly, research topics on service robots in the tourism and hospitality industry are diverse, but mainly cover four aspects: service robots, consumers, human employees, and service environment. Here, “Anthropomorphism of service robots”, “Service quality of robots”, and “Function and role of robots in the service context” belong to the research at service robot level. “Impact of service robots on consumers” and “Impact of service robots on employees” focus on the study perspective of consumers and employees, respectively. “Human–robot interaction” and “Technology acceptance of service robots” involve both consumer research and employee research. The “usage of service robots in COVID-19” is a series of studies that explore the impact of environmental factors. According to the S-D logic, service robots, consumers, employees, and service environments are vital elements of the organic system of service delivery in the tourism and hospitality industry [9], and these four components realize value co-creation through the allocation, integration, and exchange of service resources. Based on this, the content of the existing research is further summarized and analyzed around these four aspects.

#### 5.3.1. Research on the Attributes and Functions of Service Robots in Service Interaction

Service robots have acquired some human-like characteristics and abilities with the help of sensors, AI and other technologies, and their human-like attributes have attracted considerable attention from researchers. These attributes include appearance [6,28,46,50,51,52], behavior [53,54], gender [28], and language styles [30]. Anthropomorphism theory is widely used in research, which defines the tendency to attribute human characteristics and emotions to inanimate objects or animals so as to rationalize their behavior as “anthropomorphism” [55]. However, the extent of anthropomorphism of different service robots varies greatly; some do not look like humans, such as the common disc-shaped sweeping robots, and others are highly anthropomorphic, like the humanoid frontline robot in Henn-na Hotel [56]. Some researchers believe that users have a more positive attitude toward more human-like service robots [28], yet, according to the uncanny valley theory, although the public is more willing to interact with humanoid robots, an excessively high level of anthropomorphism can trigger their rejection [49], such as causing customer discomfort or disgust [50]. The same situation also occurs with employees, and studies have shown that service robots that are too human-like are more likely to make employees feel frightened and anxious about robotics [32].

Considerable research on the different types of service robots in different service environments has emerged, such as robot baristas [36], cleaning robots [57], and robot chefs [58]. The application of these service robots is generally believed to enhance the corresponding service experience for customers [59]. However, existing technologies cannot consistently guarantee the consistency and stability of robot service provision. For example, there may be poor service quality or even service failure when robots provide services to customers, which could easily lead to service dissatisfaction. These problems have further led to discussions on the difference in service quality between robots and human employees, and the failure of robot service issues. The SERVQUAL model is widely used in the study of service quality differences between robots and human employees, Chiang and Trimi [60] verified the feasibility of this model in measuring the quality of service provided by robots. Another topic, service failure, involves two different parts: failure attribution and service recovery, with the former focusing on consumers’ attribution choices for service failure liability. Studies found that consumers attributed service failures differently, and they may blame service robots [31], the service organization [61], or even themselves [31] for not being provided the appropriate level of customer service. Service recovery aims to explore effective strategies for service failure remediation, for example, humorous responses [62] and tool-based recovery strategy [63] were verified effective in robot service failure remediation.

#### 5.3.2. Research on Consumers’ Use of Service Robots and the Impact on Their Behavioral Willingness

This research branch focuses on consumer–robot interaction and behavioral willingness, the former is an important part of robots service delivery, emphasizing human-centered experience [5], and the effectiveness of this process determines the success of robot service provision [64]. The latter addresses topics such as consumers technology acceptance [51], willingness to use [56], service delivery participation [36], service evaluation [46], and service preference [65]. The issue of consumer technology acceptance is the most widely discussed, and researchers have tried to identify the determinants that affect consumer acceptance of robotic technology through one or more theories, including rational choice theory [66], planned behavior theory [67], perception value theory [37], and TAM [38]. However, current practice and research show that consumers are more likely to choose human services over robots [68]. Tussyadiah [69,70] pointed out that this phenomenon was not only due to consumers’ concern about privacy and data security but also related to technophobia, whereas Mende et al. [71] argued that these problems might relate to the humanoid characteristics of service robots that cause customers to feel threatened by identity, which in turn triggered their resistance. In addition, inefficient communication was also believed to be responsible, because the robots lack the thinking and analytical skills required during the service process, resulting in customers not being able to communicate with the robots as fluently as humans [72].

A variety of factors have been proven to affect consumers’ human–robot interaction experiences and behavioral willingness. The most discussion has occurred around three types of factors: robotic, consumer personal, and environmental factors. Robotic factors are mostly related to the functional attributes of robots, in addition to the commonly discussed anthropomorphic appearance [6,28,46,50,51,52], and also include performance efficacy [46], sociability [73], automation [51], proactivity [53], empathy [60], and competence [54]. Among consumer personal factors, the influence of demographic factors has been verified, such as gender [74], age [75], cultural background [76], education experience [77], social class [69], and previous robot service experience [78]. Researchers have also studied the psychological factors of consumers, such as trust [53,79], utilitarian motivation and hedonic motivation [80], perceived intelligence [34], perceived safety [81], perceived risk [35], perceived social presence [56], and perceived usefulness and perceived ease of use [67]. Some other scholars have pointed out that other personal characteristics of consumers also have influence, such as interaction tendency [30], political ideology [82], and information sensitivity [83]. Environmental factors like social crowding [84] and atmosphere [85] are important variables at play as well.

#### 5.3.3. Research on the Application of Service Robots by Employees and the Impact on Work Efficiency and Occupational Safety

Given that service robots can replace or assist most low-skilled human services [59], their application will inevitably have an impact on employees. Although not all positions will be replaced by robots, employees still show concerns about the application of service robots [81]. The emergence of the above problems has triggered a series of discussions on the impact of service robots on human resources in the tourism and hospitality industry. Scholars have been paying close attention to the attitudes of managers and employees toward service robots to understand the scope of their practical application. Research findings indicate that managers and employees have different views on the long-term application of service robots: managers generally have positive attitudes to the introduction of service robots because of their advantages in carrying out routine and repetitive tasks and decreasing payroll costs [86]. In contrast, employees show more resistance due to concerns about possible unemployment, with employees in positions with lower skill requirements showing stronger negative emotions [33]. 

Researchers that explored corresponding impact mechanisms believe that this is because of factors such as job insecurity [11], employees’ awareness of robotics technology [33], and technological insecurity [87], triggering negative feelings among employee. This in turn drives the reduction in employees’ job engagement and the increase in resistance to the deployment of service robots. Moreover, employees’ negative attitudes toward service robots may also lead to an increase in employee turnover [11]. The robot usage resistance model [32] provides a comprehensive understanding of the drivers of employees’ resistance to service robot usage. Additionally, researchers also discussed the acceptance of service robots by employees, among which there were studies on their usage intention [88] and willingness of human–robot collaboration [88,89] to service robots. This included variables such as perceived usefulness, perceived ease of use [88], trust [89], and perceived organizational support [33], which were shown to have a positive impact on employees’ acceptance of service robots.

#### 5.3.4. Research on the Impact of COVID-19 on the Provision and Acceptance of Robot Service

The tourism and hospitality industry is a traditional service industry with high human contact characteristics, but the outbreak of COVID-19 has limited social distancing in destinations and hotels, severely affecting service delivery. In this context, the emergence of contactless services provided by robots has addressed the urgent need for the tourism and hospitality industry to be able to provide services normally under the influence of the epidemic. Some researchers have explored the impact of COVID-19, with around 10% of empirical studies on the related topics, which were identified in the literature, retrieved and included in this paper. Most of the research focuses on the impact of COVID-19 on the use of service robots by consumers, while a small number of studies discuss the changes brought by COVID-19 to marketing and management practices of the supply side. 

According to the results, the demand side has a higher positive attitude toward service robots during COVID-19 than before, and consumers are more willing to stay at hotels and eat at restaurants with robots providing the services [75]. With the gradual normalization of COVID-19, some studies have pointed out that these changes are adjusted by the risk of the epidemic, that is, only when the risk of coronavirus transmission is high are consumers’ attitude toward robot services more positive [66]. The theories commonly used to explain these phenomena are the behavioral immune system theory, the protection motivation theory, and the health belief model, all of which explore consumer health behaviors from a motivational perspective, that is, consumers prefer robot services to protect themselves from the threat of virus from human contact. Specifically, factors such as infection cues [90] and the perceived risk of COVID-19 [35] increase customers’ willingness to use service robots [90], and positively affect their interaction with service robots [35], as well as promote customers’ reservation intention [91]. Supply side research generally uses qualitative methods to reveal the positive effects of service robots during COVID-19, such as reducing security threats from the virus, reducing companies’ fixed costs [92], and improving service supply quality [59].

### 5.4. Relation and Difference between the Four Subject Research Areas of Service Robots in the Tourism and Hospitality Industry

With the above analysis, four frontier research branches have been obtained. Given that the research topics covered in each branch are important components of the service delivery system of the tourism and hospitality industry, the contents of each section overlap around the service delivery process rather than exist independently. Specifically, studies that explore the impact of service robots’ characteristics on consumers belong to both the research on service robots and on consumers, and articles that focus on the impact of COVID-19 on the application of robots by employees not only belong to the research about employees, but are also related to research about the service environment. However, due to the different perspectives and different focus points in each branch, research on service robots pays more attention to the attributes or functions of robots. In these studies, the independent variables studied are mostly robotic factors, while the research on consumers focuses more on exploring the psychological variables of consumers, such as robot service evaluation, preference, and perception. Correspondingly, most of the research from the perspective of employees takes employee factors as the main research variable, and the related research on the topic of service environment usually takes COVID-19 as a direct or indirect impact variable to investigate the impact of COVID-19 on the provision and use of robot services. In summary, a research model was constructed to show the research frontier of service robots in the tourism and hospitality industry and reveal the inner relationship among the four research branches mentioned above (see Figure 6).

## 6. Future Research

Based on a systematic review of service robotics research in tourism and hospitality, this study provides a foundation to identify the research gaps on service robots. Follow-up research can be further deepened along the following tracks: 

(1) Increasing longitudinal data and case studies in the use of research methods would prove worthwhile. Because most of the current empirical studies use questionnaire surveys and experimental methods, current research data mostly employs cross-sectional data. There are much fewer longitudinal studies and case studies, but longitudinal studies nonetheless help to understand the changes in user attitudes and behaviors related to service robots, while case studies help to provide managers in the tourism and hospitality industry with more references for service robot applications and deployment. 

(2) Additional interdisciplinary theories must also be incorporated. The existing theory-driven orientation of research tends to use “old theories” to investigate “new questions”, making the use of theory largely fall short in innovation. Specifically, researchers generally use models or theories that are commonly used to explain human behavior in studying service robots. Some theories, such as TAM [47], may not properly represent service robots adoption. However, as robot-related technologies continue to advance, the complexity of human–robot interactions and other phenomena will further increase, and the cross-integration of interdisciplinary theories must be deepened in both content and form.

(3) The research content could be further extended. Existing studies have discussed some of the technological features of service robots, particularly their anthropomorphic appearance and the validity of the uncanny valley theory being confirmed in the extant literature. However, robot anthropomorphism is a multi-dimensional concept; in addition to appearance anthropomorphism, behavior anthropomorphism, which is assigning human behavioral characteristics (i.e., voice, action, and language style) to robots, is also an important dimension [55]. Many researchers have not strictly distinguished these two dimensions in their work. This means that questions such as “Do appearance anthropomorphism and behavior anthropomorphism have the same effect on the users’ service robots acceptance?”, “Could the uncanny valley effect also happen to service robots that do not look like humans but behave anthropomorphically?”, and “Which type of anthropomorphism makes users more willing/unwilling to accept” deserve further exploration.

Although a large number of demand-side studies have been developed, these studies are mainly concerned with the same topic, namely the consumer’s willingness to use service robots and specifically on exploring the factors that motivate consumers to accept a service robot before and during use. However, consumer behavior after using a service robot can be further investigated, such as recommendation, revisiting, and pro-environmental behavior at the service location. Furthermore, few studies seem to have explored the factors that influence consumer rejection of service robots. It is thus valuable to investigate these key points from another perspective. 

Research on the supply side should also be increased. Clearly, studies from the supply side perspective are limited in comparison to the demand-side research, with most supply side studies still merely investigating provider attitudes and views on service robots. Moreover, research topics from a supply side perspective remain scarce. Under the background of the expanding global job market and the continuous increase in the difficulty of employment, some novel problems have already emerged: Have employees’ views and attitudes toward service robots changed? What other effects will the deployment of service robots have on human resources in the service sector? Additionally, how can employee–robot collaboration be better implemented in the workplace?

As for the research on the service environment, with COVID-19 seemingly coming to an end, the threat posed by the pandemic to the public and to the economy has been significantly reduced. Although it may seem outdated to continue investigating COVID-19-related issues in the future, the presence of new phenomena and changes in robotic services in the post-pandemic era also deserves further scholarly attention.

## 7. Conclusions and Perspectives

This paper quantified and coded existing studies on service robots in the tourism and hospitality industry found in the core database of WoS using bibliometric and content analysis methods. We grouped the research themes into four areas to elucidate the latest progress and potential research gaps. The findings herein are as follows:

(1) Research work on service robots in the tourism and hospitality industry started late and has been a scholarly concern for the past decade. The outbreak of COVID-19 has also stimulated an increase in the number of studies. Currently, research outcomes show a continuous growth trend and have since attracted investment from research forces across multiple disciplines. International authoritative academic journals in the tourism and hospitality discipline contributed the most to the number of publications, with the top journal *International Journal of Contemporary Hospitality Management* publishing the most number of papers. More than 700 scholars from nearly 400 universities and colleges from different parts of the world are involved in these studies, and collaboration among them is common. Only a few, however, are high-yield researchers. Universities from China, the United States, and South Korea, where robotic technology is relatively developed, have been involved in much of the existing research.

The types of studies included reviews and empirical studies, in which quantitative, qualitative, and mixed methods were involved, although quantitative empirical research was found to be employed the most. Due to the convenience of operation, online scenario experiments and questionnaires were commonly used to obtain the data, and the processing of the data involved the mixed use of multiple analysis methods. In the use of research theory, the comprehensive application of multidisciplinary theory was most pervasive, especially the interdisciplinary integration and application of psychological theory. For example, the most frequently used TAM integrated concepts from both social psychology and information technology.

(2) Combined with service-dominant logic, the research frontier covers four main research branches, with each branch focusing on different research topics from four different perspectives, namely service robots, consumers, employees, and service environment. The contents of the four branches are related and overlap with each other around the service delivery process of the tourism and hospitality industry. Specifically, articles focusing on service robots are devoted to studying the anthropomorphism, service quality, service functions, and roles of service robots. Consumer service evaluation, preferences, and interactions are often discussed in consumer-focused studies. Studies from an employee perspective have been devoted to investigating employees’ attitudes and views toward service robots, their willingness of collaboration, and the influence of service robots on employee occupational security. Moreover, the research on the service environment has focused on the threat of COVID-19, the use of contactless services during the epidemic, and the impact of COVID-19 on cleanliness and hygiene safety in the industry. 

(3) The current research work has allowed us to identify some research gaps: on the one hand, general findings show that research methods lack longitudinal surveys and case studies, and research theories need to be more innovative. On the other hand, although the research topics are diverse, the studies on the four frontier branches (i.e., service robots, consumers, employees, and service environment) must be extended in research questions, such as the issues of the behavior anthropomorphism of service robots, the applicability of such technologies in more forms of tourism and hospitality scenarios, consumers’ rejection of service robots, the implementation of employee–robot collaboration, the impact of individuals’ experience (e.g., tourists’ travel experience or employees’ work experience) on the relationship between them and service robots, and the phenomenon of service robots application in the post-pandemic era—all of which deserve wider exploration.

## 8. Limitations

This paper reviews the academic literature on service robotics research in the tourism and hospitality industry, which is helpful to understand the latest knowledge structure and research progress in this field, but there are still some limitations that need to be improved. First, we only select the WoS database as the source of data analysis, potentially missing the excellent academic research included in other databases, resulting in the final analysis results being incomplete. Second, the literature search only obtains the research published before April 2023, and the conclusions are only a phased combing and summary of the initial stage of service robotics research in the tourism and hospitality industry. Therefore, regularly reviewing the latest research on related topics in future research is necessary so as to update the research progress in this field.

## Figures and Tables

**Figure 1 behavsci-13-00560-f001:**
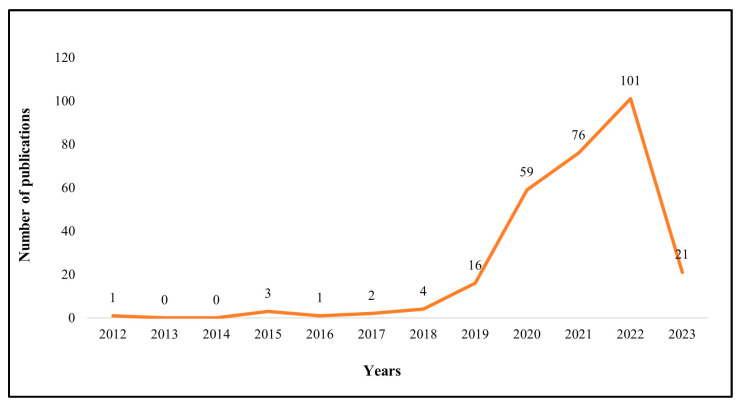
Statistics and trends of annual publications from 2012 to 2023.

**Figure 2 behavsci-13-00560-f002:**
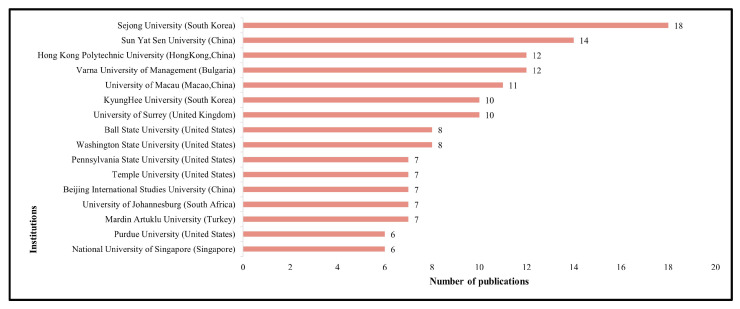
Top 16 institutions for publications.

**Figure 3 behavsci-13-00560-f003:**
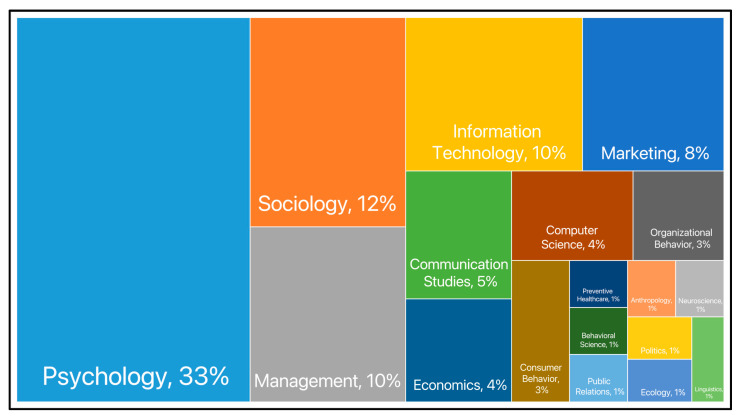
Statistics of subject area of research theory.

**Figure 4 behavsci-13-00560-f004:**
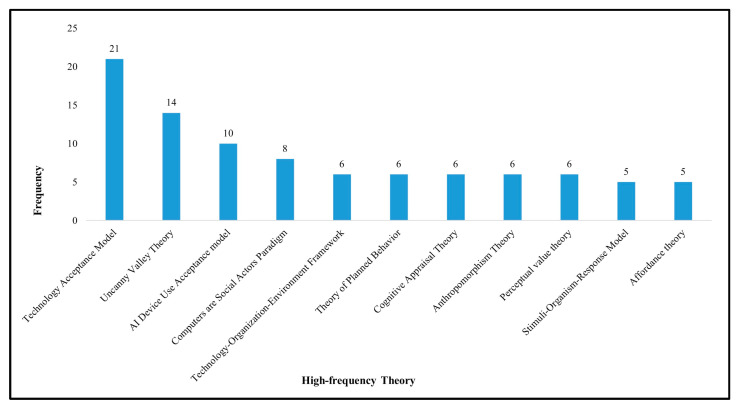
Top 11 research theories based on frequency.

**Figure 5 behavsci-13-00560-f005:**
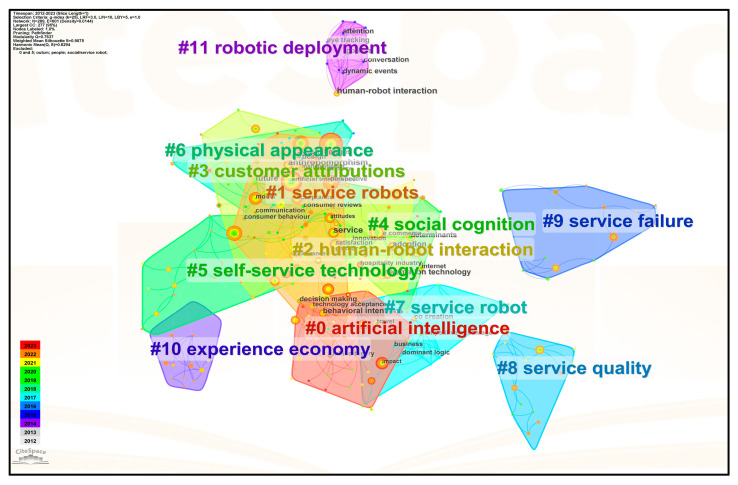
Clustering network of keywords for research on service robots in the tourism and hospitality industry ("#" represents the number sign for clustering labels, and the smaller the number after "#", the more keywords it includes in the corresponding cluster).

**Figure 6 behavsci-13-00560-f006:**
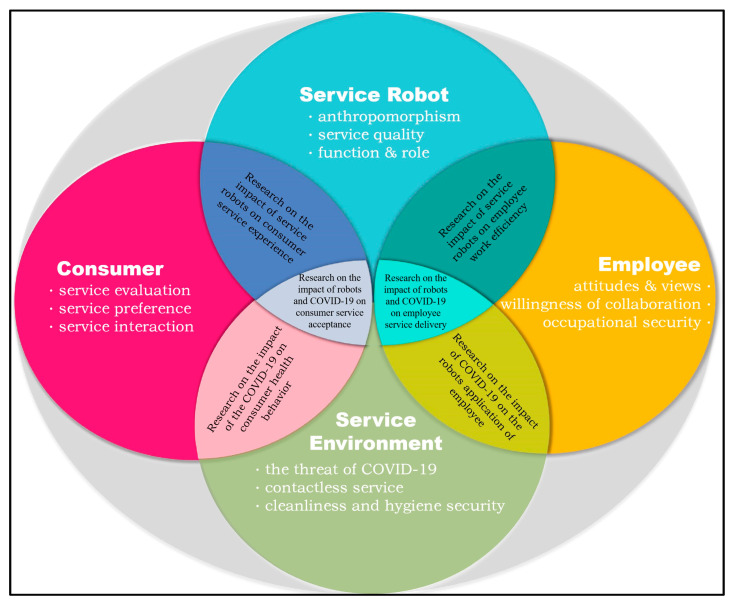
Relation between the subject research areas under S-D logic.

**Table 1 behavsci-13-00560-t001:** Summary of high-producing journals.

Journal	Publications	Category	Quartile	5YIF
*International Journal of Contemporary Hospitality Management*	37	SSCI	Q1	9.72
*International Journal of Hospitality Management*	30	SSCI	Q1	11.129
*Sustainability*	22	SCIE	Q3	4.089
*Annals of Tourism Research*	13	SSCI	Q1	13.44
*Tourism Management*	11	SSCI	Q1	13.761
*Journal of Hospitality Marketing & Management*	10	SSCI	Q1	8.025
*Journal of Hospitality and Tourism Technology*	9	SSCI	Q2	5.54
*Journal of Hospitality and Tourism Management*	8	SSCI	Q1	7.809
*Tourism Management Perspectives*	8	SSCI	Q1	7.8
*Technology in Society*	6	SSCI	Q1	6.231
*International Journal of Social Robotics*	6	SCIE	Q2	4.312
*Electronic Markets*	6	SSCI	Q2	6.378
*Computers in Human Behavior*	6	SSCI	Q1	10.097
*Tourism Review*	6	SSCI	Q1	7.003

**Table 2 behavsci-13-00560-t002:** The information of high-producing authors.

Author	Publication Frequency	Institution	Region	Research Interests
Hwang Jinsoo	13	Sejong University, College of Hospitality and Tourism Management	South Korea	service marketing, innovativeness, customer satisfaction
Stanislav Ivanov	12	Varna University of Management, School of Hospitality and Tourism Management	Bulgaria	economics of tourism, artificial intelligence and service automation, hotel marketing
Kim Heather Markham	8	Sejong University, College of Hospitality and Tourism Management	South Korea	service marketing, brand loyalty, brand attachment
Craig Webster	6	Ball State University, Department of Management	United States	robot and artificial intelligence, economics of tourism, tourism marketing and management
Tussyadiah IisP	6	University of Surrey, School of Hospitality and Tourism Management	United Kingdom	artificial intelligence, consumer behavior, sharing economy
Gursoy Dogan	6	Washington State University, Carson College Business, School Hospitality Business Management	United States	hospitality and tourism marketing, information search, services management
Seyitoglu Faruk	6	Mardin Artuklu University, Faculty of Tourism	Turkey	tourist behavior and experience, robots and robotic technology in tourism/hospitality services

**Table 3 behavsci-13-00560-t003:** Statistics of research method.

Research Method		Number	Total	N%
Quantitative Research	questionnaires	88	169	60%
	experiments	71		
	experiments and questionnaires	6		
	others	4		
Qualitative Research	literature analysis	23	80	28%
	interview	20		
	data mining	17		
	case analysis	5		
	focus group discussion	3		
	others	12		
Mixed Methods Research	questionnaires and interview	15	35	12%
	bibliometric and content analysis	8		
	experiments and interview	3		
	questionnaires and observation	2		
	others	7		

**Table 4 behavsci-13-00560-t004:** Classification and coding table of keywords in the research of service robots in tourism and hospitality industry.

Code	Category	Subcategory
G1	Technology acceptance of service robots	artificial intelligence, technology acceptance, information technology, adoption intention, user resistance, performance expectancy, user experience, technology readiness, technology self-efficacy
G2	Usage of service robots in COVID-19	social distancing, COVID-19 pandemic, risk perception, public health emergency, cleanliness, social withdrawal tendency
G3	Human–robot interaction	human–robot interaction, meaningful engagement, tourist engagement, tourist–robot interaction, service encounter, employee–robot collaboration, client interaction, value co-creation
G4	Impact of service robots on consumers	brand loyalty, brand satisfaction, brand attitude, purchase intention, motivated consumer innovativeness, consumer behavioral intention, compensatory consumption, word-of-mouth intentions, brand experience, customer features, demand-side perspective
G5	Anthropomorphism of service robots	physical appearance, uncanny valley theory, empathetic intelligence, robotics awareness, anthropomorphic service robots, identity threat, robotic empathy, gender stereotypes
G6	Service quality of robots	service quality, service failure, instrumental recovery, informational recovery, customer attributions, experience quality
G7	Function and role of robots in the service context	robot barista, service proactivity, robotic restaurants, frontline robots, robot concierge, robotic chef, social presence, social robot, autonomous robots, service automation, semi-automation service, task orientation
G8	Impact of service robots on employees	robotic deployment, service employees, employee creativity, employee competence, service robot risk awareness, manager perceptions, turnover intention, job engagement, job insecurity, supply side perspective

## Data Availability

Data are included in this manuscript.

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
