# Peer review of "Research on the Frontier and Prospect of Service Robots in the Tourism and Hospitality Industry Based on International Core Journals: A Review"

_behavsci, 2023, doi:10.3390/bs13070560_

Round 1

Reviewer 1 Report

The issue of the impact caused by service robots has aroused academic attention. One can certainly see a large increase in studies examining the application of service robots in tourism and hospitality, focusing primarily on their impact on consumers, employees, and service quality.

Interesting choice to deepen and explore a subject where the amount of scientific literature is very scarce at the moment.

Methodologically, the quantitative, qualitative and mixed use is correct for an approximation to reliable results.The limitations of the study would be the use of a single database such as WOS, leaving aside other prestigious ones, rectifiable in future research.

Improve research, since it is not clear if customers prefer a robot service or human treatment.

Author Response

Dear Reviewer:

We would like to express our sincere appreciations of your valuable comments concerning our paper. Regarding your concerns about the content of the paper, we have made careful consideration and provide a point-to-point response. Please see the attachment.

Reviewer 2 Report

In the paper titled, Research on the frontier and prospect of service robots in the tourism and hospitality industry based on international core journals: a review, author/s present comprehensive and systematic review of existing research regarding service robots in the tourism and hospitality industry, using papers and studies referenced in WoS, providing insights for future studies, using bibliometric and content analysis methods (based on the S-D logic via content analysis combined with cluster analysis of CiteSpace). However, it does not introduce anything that stands out as novel or different from the insights presented by the papers referenced. Despite this, the narration and organisation of the paper is extremely informative and could be used in teaching processes at the university level in the field of new technologies in tourism.

However, this paper requires reflection in accordance with the comments that came to my mind after reading it. When reading the text of the article, I noticed a few aspects that could improve the proposed concept. These are of minor significance, and I encourage the author’/’s to introduce these minor revisions. I believe that the elements outlined in the comments are important, and addressing them would help dispel any doubts related to the findings of the study. I sincerely hope you will find these helpful in improving your manuscript.

Some thoughts, just for consideration:

-          The parts of this paper are extremely valuable: 4.4. Research methodology in publications and 4.5. Research theory showing a broad range of attempts to analyse the use of robots in hospitality and tourism as well as theories from managerial point of view and economic or social sciences, explaining those relations. The paragraph 6. Future research provides a very solid foundation to identify the research gaps on service robots and is crucial for this analysis. For the purpose of elucidating the latest developments and possible research gaps, this section groups the research themes into four areas. That summary is enough critical, as also the previous parts of the paper should be. But, the graphic material illustrating those issues, like Figure 2, 3 and Figure 4 are not reader-friendly.

-          The literature, that author/s refer to, should be more critical and less narrative. You can show that some specific studies would show better results using the methodologies discussed or any other method/models in your review.

-          I suggest, for consideration, to name the last part of the article, titled 'Conclusion' as: Conclusion and outlook / perspectives. It might be more useful.

-          In the conclusion part, the author/s describe: “Specifically, articles focussing on service robots are devoted to studying the anthropomorphism, service quality, service functions, and roles of service robots. Consumer service evaluation, preferences, and interactions are often discussed in consumer-focused studies. Studies from an employee perspective have been devoted to investigating employees' attitudes and views towards service robots, willingness of collaboration, and the influence of service robots on employee occupational security.”. Open questions could be asked if such technologies are applicable to be used in many more forms of tourism and how individualisation of tourism experience will change the relation between tourists and robots?

-          This paper needs some touch of editing, because there are some paragraphs justified on both edges and some only on the left.

Author Response

Dear Reviewer:

Thanks very much for  your generous comments and insightful suggestions.  We have read through comments carefully and have made revision. We hope that these modifications will meet with your approval. Please see the attachment.

Reviewer 3 Report

1. The authors do not explain why the Scopus base is also not considered. I recommend either considering it also, or at least explaining its non-consideration and mention

2. About the software used - it is advisable to mention whether it is free, paid, the source of its receipt, the legitimacy of use and the possibility of use by readers as well

3. ALL Figures - the texts are VERY too small, unreadable, it is better to enlarge for visibility

4. 

Author Response

Dear Reviewer:

Thank you for your valuable and helpful comments on improving the quality of our manuscript. We have made some modifications in the revised manuscript. We hope that these corrections will meet with your approval. Please see the attachment.
